# Giant linear strain gradient with extremely low elastic energy in a perovskite nanostructure array

Y.L. Tang[1,*], Y.L. Zhu[1,*], Y. Liu[1], Y.J. Wang[1] & X.L. Ma[1,2]

Although elastic strains, particularly inhomogeneous strains, are able to tune, enhance or create novel properties of some nanoscale functional materials, potential devices dominated by inhomogeneous strains have not been achieved so far. Here we report a fabrication of inhomogeneous strains with a linear gradient as giant as $10^6$ per metre, featuring an extremely lower elastic energy cost compared with a uniformly strained state. The present strain gradient, resulting from the disclinations in the $BiFeO_3$ nanostructures array grown on $LaAlO_3$ substrates via a high deposition flux, induces a polarization of several microcoulomb per square centimetre. It leads to a large built-in electric field of several megavoltage per metre, and gives rise to a large enhancement of solar absorption. Our results indicate that it is possible to build up large-scale strain-dominated nanostructures with exotic properties, which in turn could be useful in the development of novel devices for electromechanical and photoelectric applications.

[1] Shenyang National Laboratory for Materials Science, Institute of Metal Research, Chinese Academy of Sciences, Wenhua Road 72, Shenyang 110016, China. [2] School of Materials Science and Engineering, Lanzhou University of Technology, Langongping Road 287, Lanzhou 730050, China. * These authors contributed equally to this work. Correspondence and requests for materials should be addressed to X.L.M. (email: xlma@imr.ac.cn).

When an elastic strain is exerted on a nanostructure, some properties of the material are tuned, enhanced or created[1–13]. For example, straining Si channels by incorporating SiGe sources and drains boosts the performance of silicon transistors without aggressively scaling their dimensions[13,14]. Moreover, inhomogeneous strains are able to break the symmetry of a nanoscale crystal and consequently give rise to some exotic phenomena in the material's properties[15]. It is known that a nonuniform strain is able to continuously tune the bandgaps of a semiconductor, enhance the optoelectronic and energy-harvesting efficiencies for $MoS_2$ monolayer and ZnO microwires[3,16,17], and induce novel room temperature metal–insulator transition in $VO_2$ nanobeams[18]. Particularly, through flexoelectric/piezoelectric couplings[15], bending strain gradients enable to induce polarization in paraelectric $SrTiO_3$ cantilever actuators[19,20]. Furthermore, strain gradient may also induce cation non-stoichiometry and Cottrell atmospheres around dislocation cores as reported in perovskite ferroelectric films[21].

It has been proposed that novel devices based on a built-in strain gradient would be of technologically importance. These should involve at least the lead-free piezoelectric elements containing no piezoelectric materials working through flexoelectric couplings[22–29], perovskite based photovoltaics/photocatalysis[30,31], flexible devices[32] and gradient functional materials[33].

Nevertheless, the above proposals are based on either the theoretical simulations[16,24,25,27] or mechanical bending of nanofibers/cantilevers[17–20] or limited information near the domain walls of ferroelectrics[30,31]. Although mechanical buckling seems an effective way to pattern stretchable semiconductor and ferroelectric nanoribbons[32,34], the corresponding attainable strain gradients are relatively lower and failure tends to occur when manipulating brittle ceramics such as ferroelectrics. Two preconditions must be met toward making use of inhomogeneous strains. The first is to find an approach of assembling inhomogeneous strains into a nanostructure; and the second is to make these complex strains precisely measured. To quantitatively measure an inhomogeneous strain is always a great challenge, since peak broadening in diffraction experiments results from not only inhomogeneous strains but also a number of other components such as crystallographic defects, refined grains and similarly orientated domains[35].

In this work, we report a fabrication of giant linear strain gradients in the lead-free perovskite nanostructures and their real-space measurements at the atomic scale. We choose single-phase room temperature multiferroic $BiFeO_3$ as a perovskite prototype[36–38]. The strain extraction is based on the geometric phase analysis (GPA, refs 39,40) technique. A special array of $a<100>$ and $a<110>$ type interfacial dislocations is identified to account for the lattice mismatch of the heteroepitaxial system. Such a relaxation mechanism results in a long-range and giant linear strain gradient across the rhombohedral (R phase) $BiFeO_3$ nanostructures displaying remarkable disclination characters. We find that the giant strain gradient enables to transfer when the $BiFeO_3/LaAlO_3$ films are repeatedly grown, suggesting a controllability of such inhomogeneous strains in abundant lead-free perovskites without the need of taking dimension constrain into account. Our results deviate from the general understanding that giant strain gradients and the resultant flexoelectricity only matter at a nanoscale, and provide an example to quantitatively measure the inhomogeneous strains through directly atomic imaging.

## Results

### Preparation of LaAlO₃/BiFeO₃ nanostructures.
Self-assembled nanostructures of $LaAlO_3/BiFeO_3$ (LAO/BFO) are epitaxially grown on a LAO(001) substrate by pulsed laser deposition with a high deposition flux mode. High frequency 5 Hz is chosen and a fresh surface of the ceramic target (mechanically polished) is used to insure a high growth rate of the first BFO layer. In the further growth of upper LAO and BFO layers, the laser was first focused on the ceramic targets for 30 min pre-sputtering to stabilize the target surfaces. The stabilized targets allow a lower deposition flux (detailed in Supplementary Note 1). The objective of the BFO/LAO/BFO film design here is to study the strain interactions and transfers among the lead-free multilayer structures and corresponding strain induced physical effects.

**Observation and strain measurements.** High-angle annular dark-field (HAADF) scanning transmission electron microscopic (STEM) imaging is applied to investigate the atomic details. Figure 1 shows an atomic morphology of a LAO/BFO/LAO(001) nanostructure. The high deposition flux in the present study avoids the 2D growth of BFO film and thus self-assembled 3D nanostructures are fabricated (more is shown in Supplementary Figs 1 and 2). For a comparison, a 2D growth of BFO film on LAO substrate under low deposition flux mode is displayed in Supplementary Fig. 3, where the famous strain-driven tetragonal BFO phase (T phase) is observed and the BFO/LAO interface is free of interfacial dislocations. The intensity of HAADF–STEM image is strongly dependent on the atomic number of corresponding element (proportional to $\sim Z^2$, detailed in Supplementary Note 2)[10,41,42]. The interface between BFO and LAO substrate is clearly seen because of respective atomic number of heavy Bi (83) and lighter La (57) atoms. Interfacial defects are identified at interfaces as labelled with pink and green arrows which indicate different types of misfit dislocations. The dislocation arrays are displayed in a low magnification image shown in Supplementary Fig. 4. To reveal the features of the defects and their effects on the BFO nanostructures, four areas each of which is labelled with a rectangle in Fig. 1a are magnified and seen in Fig. 1b–e, respectively. Compared with the LAO substrate, the lattice of BFO near the left area 1 is parallel to the LAO lattice (Fig. 1b), while the lattice of BFO near the right area 2 obviously rotates in a counterclockwise fashion (Fig. 1c). The rotation angle in 2 is $\sim 4°$. The interfacial defect is composed of two kinds of interfacial dislocations whose Burgers vectors is $a[010]$ and $a[011]$, respectively, based the Burgers circuits drawn in Fig. 1d,e. Here $a$ indicates the lattice parameter of a cubic perovskite unit-cell. Thus the green and pink arrows in Fig. 1a indicate $a[010]$ and $a[011]$ dislocations, respectively. Both of the dislocations contain $a[010]$ Burgers vector components, which relax the large mismatch between BFO ($a = 3.96$ Å) and LAO ($a = 3.79$ Å, ref. 7). It is proposed that the high deposition flux in the present study triggers the formation of dislocations other than an intergrowth of R and T (tetragonal) phases to relax the mismatch. As is seen in Fig. 1e, the $a[001]$ component of the $a[011]$ dislocation has a giant effect which makes a rotation of the BFO lattice. However, the strain effect of a single dislocation is extremely local, thus the lattice rotation in the BFO (area 2) away from the interface is actually derived from a synergetic effect of the special array of interfacial dislocations.

GPA and directly atomic position mappings are effective and accurate tools which show great potential to extract complex and inhomogeneous strain distributions[8,10,39,40,43–45]. We apply GPA to extract the dislocation configuration and unique strain distributions in the nanostructures since GPA works better for long-range strains[40]. Figure 2a,b are the lattice rotation ($\omega$) and in-plane strain ($\varepsilon_{xx}$) maps of the LAO/BFO/LAO nanostructure. Giant lattice rotation resulting from the interfacial dislocations array is obviously visible. The lattice rotation map exhibits

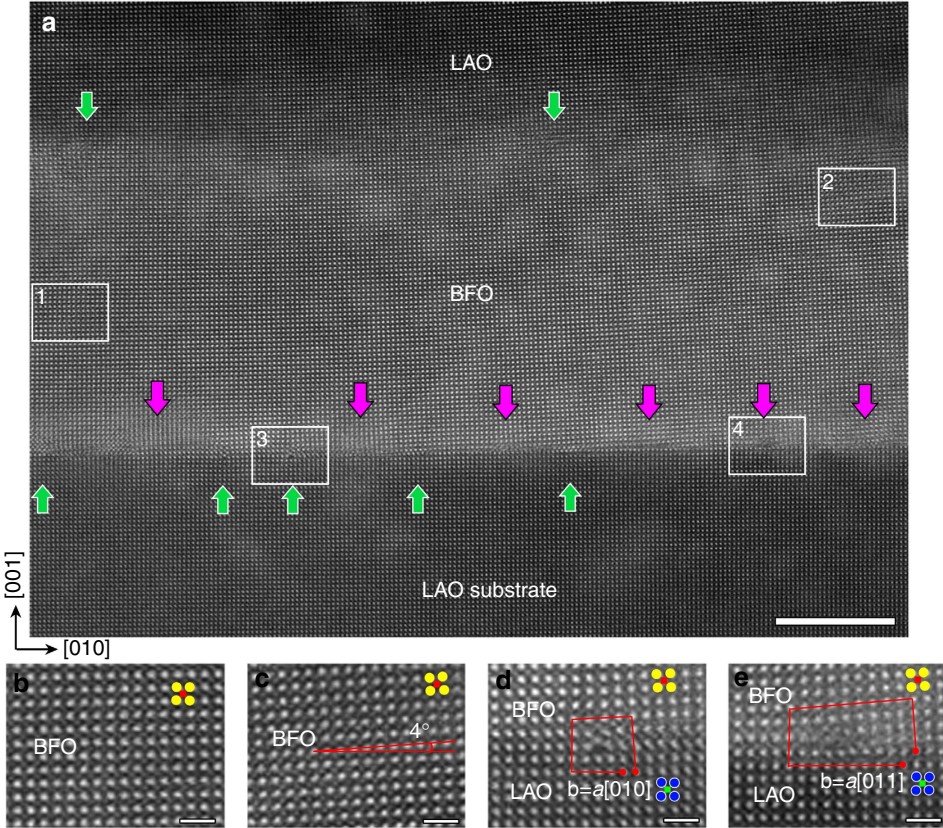

**Figure 1 | HAADF–STEM images of the two-layer LaAlO₃/BiFeO₃/LaAlO₃(001) nanostructures viewed along [100].** (**a**) Atomically resolved HAADF–STEM image. Scale bar, 10 nm. Boxes labelled with numbers 1, 2, 3 and 4 are four typical areas magnified in **b**–**e**. Note the relative lattice rotation of **c** compared with **b**, which indicates a possible continuous lattice rotation in the BiFeO₃ nanostructures in **a** across the in-plane direction. Pink and green arrows in **a** indicate two different types of interfacial dislocations, as shown in **d** and **e**. The insets in **b**–**e** correspond to the unit-cell schematics of BiFeO₃ and LaAlO₃ (yellow circles, Bi; red squares, Fe; blue circles, La; green squares, Al; O is omitted here since it was not imaged because of its lower atomic number). All scale bars in **b**–**e** are 1 nm.

systematical inhomogeneity, which increases gradually from left to right of the image. Three line-profiles of the lattice rotations are shown in Fig. 2c. The continuous lattice rotation in both BFO and top LAO nanostructures can be clearly seen, which is reminiscent of bending distortion of crystal lattice. Bending deformation must accompany a lateral deformation[46]. To gain further insight on the long-range strain distribution (lateral deformation) in the nanostructures, in-plane strain ($\varepsilon_{xx}$) map was extracted, as shown in Fig. 2b. The dislocation type is reconfirmed via the comparisons of in-plane ($\varepsilon_{xx}$) and out-of-plane ($\varepsilon_{yy}$, Supplementary Fig. 5) strain maps. The Burgers vector of $a$[010] dislocations possesses only in-plane component, while $a$[011] dislocations have both in-plane and out-of-plane components. Thus the dislocation cores displaying only out-of-plane strain contrast are $a$[011] dislocations (as marked by pink arrows in Fig. 2 and Supplementary Fig. 5), and the others are $a$[010] dislocations (marked by green arrows).

It is worthwhile to mention that the array of these dislocations is in an aperiodic fashion (Fig. 2). At the left side, two $a$[011] dislocations are separated by three $a$[010] dislocations; in the middle, $a$[010] and $a$[011] dislocations appear one by one; and at the right side, only $a$[011] dislocations emerge. The $a$[011] dislocation contains out-of-plane component ($a$[001]) and consequently rotates the BFO lattice. However, at the left side, the $a$[011] dislocations are well separated by the $a$[010] dislocations which contain no out-of-plane component, so their rotation effect is restrained. Nonetheless, at the right side, the three well aligned $a$[011] dislocations result in giant lattice

rotation in the BFO and LAO lattice, which spreads for a long distance across the entire BFO and LAO films. Finally, the right BFO lattice is severely rotated, while the left BFO lattice still holds the same orientation as the LAO substrate. This synergetic effect of the $a$[001] components resembles the low-angle boundaries (or tilt boundaries) formed by $a$[001] dislocation array in perovskite bicrystal[47]. In fact, the present situation corresponds to a concept of partial disclination, which is responsible for the giant long-range lattice rotation and behaves different from the well known a[100] dislocation arrays[48].

As above mentioned, continuous lattice rotation is relevant to a bending deformation, thus the lateral deformation must be accompanied[46]. The $\varepsilon_{xx}$ strain in the LAO/BFO/LAO lattice show obvious and systematic inhomogeneity, as shown in Fig. 2b. The $\varepsilon_{xx}$ decreases from the bottom to top, which is consistent with the character of bending deformation of materials[46]. A quantitative line-profile of $\varepsilon_{xx}$ is shown in Fig. 2d. The strain gradients of $\varepsilon_{xx}$ can be seen directly as the slopes of strains in the BFO and above LAO lattice. Note the strains in the LAO substrate reference lattice are constant 0 which confirms the validity of strain gradients in above BFO and LAO lattice. The strain gradients of $\varepsilon_{xx}$ in BFO are estimated by the slopes of the curves, which are well above $10^{6}$ m$^{-1}$ order. Such a giant linear strain gradient is not occasionally observed in the present lead-free perovskite and more examples are displayed in Supplementary Figs 6–11 and Supplementary Note 3.

The giant linear strain gradient identified in the present study can be preserved and transferred into multilayers. For example,

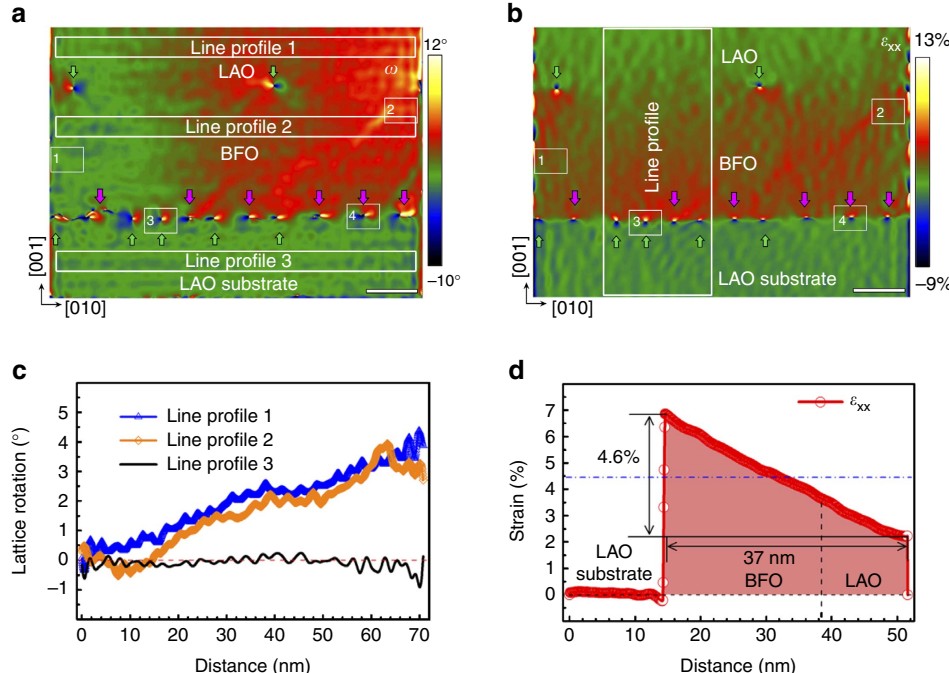

**Figure 2 | Lattice rotation ($\omega$) and in-plane strain ($\varepsilon_{xx}$) maps of the two-layer LaAlO$_3$/BiFeO$_3$/LaAlO$_3$(001) nanostructures.** (**a**) 2D lattice rotation ($\omega$) and (**b**) in-plane strain ($\varepsilon_{xx}$) maps via GPA. LaAlO$_3$ substrate is chosen as the reference lattice. Scale bars in **a** and **b** are 10 nm. Note the obvious contrast at the dislocation cores and the continuous change of lattice rotation in both LaAlO$_3$ and BiFeO$_3$ layers across the in-plane direction. Boxes labelled as 1, 2, 3 and 4 are four typical areas correspond to that in Fig. 1a. Profiles corresponding to the three marked lines in **a** are visualized in **c**. The continuous increase of lattice rotation in both BiFeO$_3$ and LaAlO$_3$ nanostructures can be clearly seen, which changes from 0° (left) to $\sim$4° (right). Note the transfer of lattice rotation from the BiFeO$_3$ to the top LaAlO$_3$ nanostructure. Also note the continuous decrease of in-plane strain in both LaAlO$_3$ and BiFeO$_3$ nanostructures across the out-of-plane direction. A white boxed area labelled as line profile in **b** is chosen as a visualization line-profile shown in **d**. A blue dotted line in **d** indicates the nominal mismatch magnitude (4.5%) for BiFeO$_3$/LaAlO$_3$(001). The strain gradient of $\varepsilon_{xx}$ is estimated by the slopes of the curve in **d**, which is well above $10^6$ m$^{-1}$ order.

when BFO nanostructure is further grown on top of the LAO/BFO/LAO(001) nanostructures with lower growth rate (Fig. 3, detailed in Supplementary Note 1), the bending deformation and the giant strain gradient are transferred into the top BFO islands (Fig. 4a–c). All the top LAO/BFO interfaces (except for the BFO/LAO(001) substrate interface) are relaxed only by $a$[010] type dislocation arrays, consequently the bending of all above perovskite lattice is preserved, since there is no $a$[001] component to accommodate the lattice rotation (Fig. 4a). The out-of-plane ($\varepsilon_{yy}$) and shear ($\varepsilon_{xy}$) strain mappings are given in Supplementary Fig. 12. Thus our results suggest a practical approach to integrate such linear strain gradient into perovskite nanoislands without taking into account of the thicknesses of these structures and magnitude of mismatches (as schematically illustrated in Fig. 4d). Our results deviate from the general understanding that giant strain gradients and the resultant flexoelectricity only matter at a nanoscale. The utility of such giant strain gradients can thus be extended as novel gradient functional nanostructures.

## Discussions

The formation of linear strain gradient at nanoscale can be understood through an elastic energy consideration, and we find that the elastic energy consumption for producing such a giant strain gradient is unexpectedly low, as shown in Fig. 5a. We consider a BFO nanoisland with dimension $l \times t \times h$, where $l$ and $t$ are the in-plane dimensions vertical to or along the imaging direction (Fig. 1a), $h$ is the out-of-plane thickness. To compare the elastic energy of a BFO island in a fully 2D strain state with that in a linear strain state, we arbitrarily set $l = t = 100$ nm in our

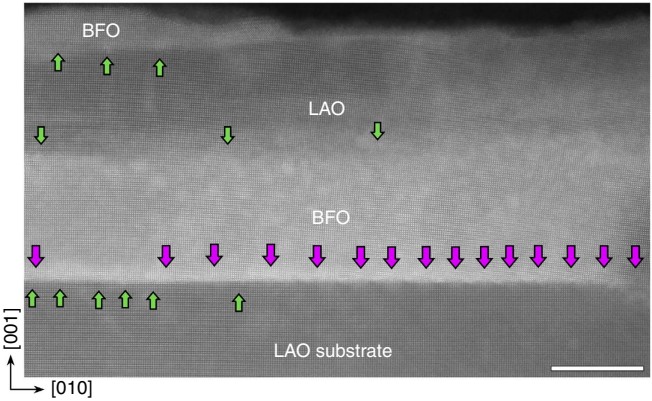

**Figure 3 | Atomic resolved images of a BiFeO$_3$/LaAlO$_3$/BiFeO$_3$/LaAlO$_3$(001) multilayer island.** Scale bar, 20 nm. The first BiFeO$_3$ layer was deposited under a high deposition flux mode. Note that there are only $a$[011] type dislocations at the right side, while the $a$[010] type dislocations dominate at the left side, as marked with pink and green arrows, respectively. This dislocation configuration suggests that giant lattice rotation and resultant giant strain gradient may occur in the BiFeO$_3$/LaAlO$_3$/BiFeO$_3$ multilayer nanostructure.

calculations. The detailed deducing process is shown in Supplementary Note 4 and Supplementary Figs 13 and 14. The elastic energies for linear strained and fully strained BFO islands are plotted versus $h$ as curve 1 and 4 in Fig. 5a. The energy of creating interfacial dislocation arrays of a 100 nm $\times$ 100 nm BFO/LAO(001) interface is plotted as curve 2, which is calculated

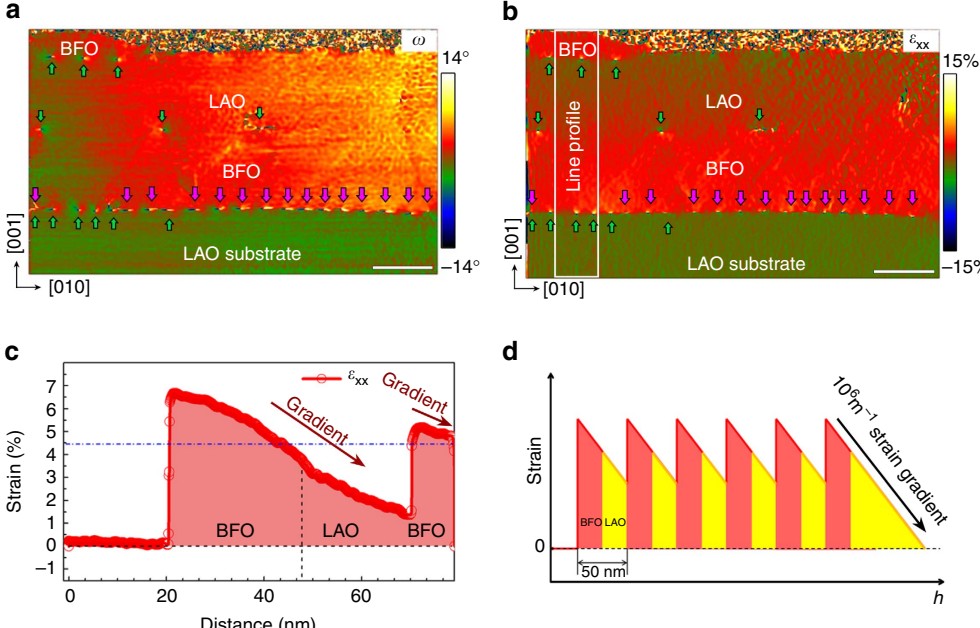

**Figure 4 | Continuous strain gradient in the multilayered BiFeO₃/LaAlO₃/BiFeO₃/LaAlO₃(001) nanostructures.** (**a**) 2D lattice rotation map ($\omega$), and (**b**) in-plane strain ($\varepsilon_{xx}$) map via GPA. Scale bars in **a** and **b** are 20 nm. Note the continuous change of lattice rotation all over the BiFeO₃/LaAlO₃/BiFeO₃ multilayer nanostructure across the in-plane direction. A white boxed area labelled as line profile in **b** is chosen as a visualization line-profile as shown in **c**. The resultant continuous decrease of in-plane strain in the first LaAlO₃/BiFeO₃ double-layer across the out-of-plane direction is remarkable. Note that except for the BiFeO₃/LaAlO₃(001) substrate interface, the two BiFeO₃/LaAlO₃ interfaces are relaxed only by $a$[010] dislocations, thus the bending deformations transfer to each layer above the substrate, since there is no other $a$[001] components to relax the bending deformations. Note that giant strain gradient also transfers to the top BiFeO₃ layer. Thus the linear strain gradient can be patterned into potential device elements by simply a multilayer growth of various kinds of perovskite nanostructures, without concerns about the dimensions and mismatches. (**d**) A schematic illustration for the preservation of linear strain gradient in perovskite nanostructures through a multilayer strategy. This strategy allows a periodic recovering of in-plane strains which do not severely increase the elastic deformation of each BiFeO₃/LaAlO₃ bilayer.

through the elastic properties of dislocations[49]. Thus the whole energy for producing the observed linear strain gradient is plotted versus $h$ as curve 3 (through $1+2$). The elastic energy of the fully strained BFO island displays highly increasing linear distribution with increasing of thickness, which is calculated through the elastic properties of 2D strained films[50] by considering the mismatch (4.5%) between BFO and LAO. It is seen that the elastic energy cost for the linear strain is so negligible that it is even much lower than the dislocation energy, especially when the thickness of the BFO island is small (note misfit dislocations in epitaxial perovskite oxides are commonly seen). Thus the observed giant strain gradients here are elastically feasible. A schematic illustration of the experimentally observed giant strain gradients in LAO/BFO islands is given in Fig. 5b.

The present study features general implications for other perovskite oxides. We further formulate the elastic energy consumption distribution versus thickness ($h$) and the location of neutral plane ($y$) of the BFO nanostructure under linear strain gradient, as shown in Fig. 5c. By considering the observed strain gradient ($\sim 1.24 \times 10^6$ m$^{-1}$) and maximum elastic limit of nanomaterials ($\sim 10\%$, ref. 51), the possible maximum $h_m$ is $\sim 160$ nm ($h_m = 2 \times 10\%/(1.2 \times 10^6$ m$^{-1}) \approx 160$ nm, when the neutral plane $y$ is in the middle of $h_m$). Note that the minimum elastic energy consumption occurs when $h = 2y$, that is, pure bending (more details can be found in Supplementary Note 4). The elastic energy consumption distribution versus thickness ($h$) and mismatch of the same size BFO nanostructure under fully 2D strained state is shown in Fig. 5d. By comparing Fig. 5c,d, it is obvious that for large mismatch systems with small thickness, the BFO nanostructure under linear strain gradient exhibits negligible elastic energy consumption compared with the fully 2D strained

state. For instance, for an $h = 50$ nm BFO nanostructure under linear strain gradient, its elastic energy is probably $<1 \times 10^{-14}$ J. However, when the same BFO nanostructure is fully 2D strained on a LAO substrate, its elastic energy is $\sim 13 \times 10^{-14}$ J, which is more than 10 times bigger than the former ones. The elastic energy consumption for linear strain state tends to more negligible when the thickness $h$ decreases.

It is known that the 2D compressive strain strongly determines the domain structures, phase constitutions and piezoelectricity. And many unique properties of BiFeO₃ films grown on substrates with smaller lattice parameters, like LaAlO₃ (refs 1,2,7,37,52,53), have been found. Except for the well-known strain-driven morphotropic-phase-boundary mechanism[7], our results reveal a peculiar mismatch relax mechanism in the large mismatch systems via the well aligned alternating array of $a<100>$ and $a<110>$ interfacial dislocations. Here the mismatch relax mechanism is dominated by mechanical property (elastic limit) of perovskite materials other than the generally accepted elastic energy considerations of epitaxial films[50,54]. Particularly, this mismatch relax process allows patterning giant linear strain gradient and the resultant flexoelectricity in abundant lead-free perovskite oxides.

The giant linear strain gradient over $10^6$ m$^{-1}$ (defined as $\Delta S$) corresponds to a radius of curvature ($r$) much $<1,000$ nm, which is probably the biggest gradient attainable in long-range linear strains expect for the local strain gradients observed at ferroelectric domain walls and other interfaces[30,31]. Experimentally, it is known that the measured flexoelectric coefficients ($f$) for perovskite are on the order of $10^{-9} - 10^{-8}$ Cm$^{-1}$ (refs 19,20,30). The present giant strain gradient over $10^6$ m$^{-1}$ may induce several $\mu$Ccm$^{-2}$ flexoelectric polarizations ($P_f$) as estimated

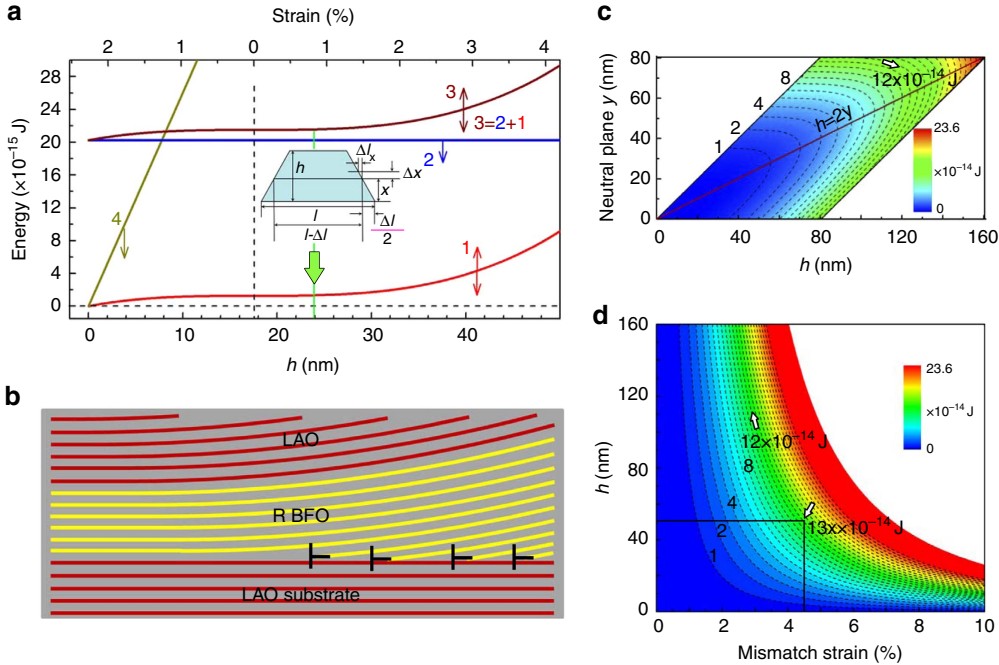

**Figure 5 | Low elastic energy states for BiFeO₃ disclinations. (a)** Comparisons of elastic energies for BiFeO₃ nanostructures under linear strain gradient and homogeneous 2D strain. The inset in **a** is a schematic indicating the elastic energy calculation involving the strain gradient. Curve 1 is the elastic energy distribution of a BiFeO₃ nanostructure under linear strain gradient. Curve 2 is the interfacial dislocation energy of the BiFeO₃ nanostructure. Curve 3 is the whole energy of the BiFeO₃ nanostructure calculated through 1+2, which represents the energy consumption for producing the experimentally observed disclinations shown in Fig. 1. Curve 4 is the elastic energy of a same sized BiFeO₃ nanostructure under fully 2D strain on the assumption that the BiFeO₃ nanostructure is fully strained by the mismatch ($\sim$4.5%). The black dotted line indicates the neutral plane where the BiFeO₃ lattices are under free strain. The green arrow indicates the thickness termination ($\sim$24 nm) of the present BiFeO₃ nanostructure (Fig. 2d). Note that, compared with the fully 2D strained state, the present observed disclination strain states exhibit almost negligible elastic energy consumptions, especially when the thickness of the nanostructure tends to smaller. A schematic illustration of the disclination formed through interfacial dislocation arrays is shown in **b**. The elastic energy consumption distribution versus thickness ($h$) and the location of neutral plane ($y$) of the BiFeO₃ nanostructure under linear strain gradient is shown in **c**. The elastic energy consumption distribution versus thickness ($h$) and mismatch of the same size BiFeO₃ nanostructure under fully 2D strained state is shown in **d**. By comparing **c** and **d**, it is obvious that for large mismatch systems with small thickness, the BiFeO₃ nanostructure under linear strain gradient exhibits negligible elastic energy consumption compared with the fully 2D strained state.

according to $P_f = f \times \Delta S$. Although this value is not so large compared with the spontaneous polarization ($P_s$) of BFO, an important signification is that the flexoelectric behaviour under the form of giant linear strain gradient is applicable for all dielectric materials when integrated into an electronic devices.

In addition, the flexoelectric coupling also leads to a large flexoelectricity-driven electric field ($E_f$), which can be estimated according to the following formula[29]:

$$E_f = \frac{e}{4\pi\varepsilon_0 a} \Delta S \qquad (1)$$

where $e$ is the electronic charge ($1.602 \times 10^{-19}$ C), $\varepsilon_0$ is the permittivity of free space ($8.854 \times 10^{-12}$ Fm$^{-1}$), $a$ is the lattice parameter. An $E_f$ well exceeding 1MVm$^{-1}$ is estimated. This value is comparable to the internal field in the conventional p–n junctions and Schottky diodes[29,55]. Details in terms of how such a large field affects the electronic properties of epitaxial interfaces and its couplings with other order parameters are still open questions. Future studies on BiFeO₃ and other perovskite materials nanostructures as flexible electronics, electromechanical or photoelectric devices could thus be stimulated. It is worthwhile to mention that, although flexoelectricity is generally discussed in terms of dielectric insulators, a very recent study by a three-point bending reveals that the flexoelectric-like coupling is much larger in doped oxide semiconductors than in dielectric insulators[56]. Thus, we propose that, by using doped lead-free perovskite oxides, it is possible to

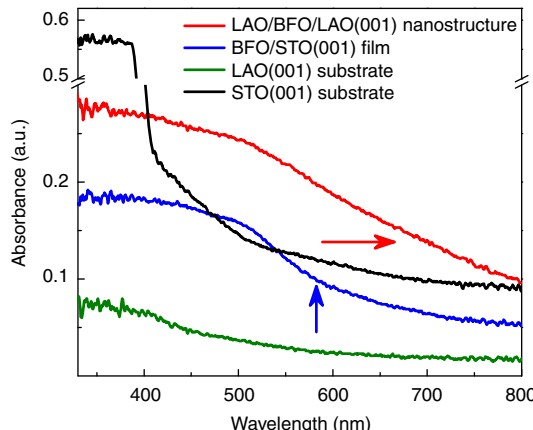

**Figure 6 | Ultraviolet–visible absorption measurements on the LaAlO₃/BiFeO₃/LaAlO₃(001) nanostructures.** A uniform BiFeO₃/SrTiO₃(001) film with $\sim$30 nm thickness, a bare LaAlO₃(001) substrate and a bare SrTiO₃(001) (STO(001)) substrate are also measured here for comparison.

construct nanostructures with giant linear strain gradient where the flexoelectric-like behaviours could be further enhanced for electromechanical applications.

To further verify how the strain gradient affects the macroscopic property of the present perovskite nanostructures, we have

performed ultraviolet–visible absorption measurements on the multilayered LAO/BFO nanostructures (Fig. 6). For comparison, a uniform BFO/STO(001) film with nearly the same thickness as the BFO layer in the nanostructure, a bare LAO(001) substrate and a bare STO(001) substrate were also comparatively measured. First we can see that the absorption of pure LAO(001) substrate is weak since the electronic structure of Al is insensitive to ultraviolet–visible excitations. In contrast, the pure STO(001) substrate exhibit obvious absorption edge (400–450 nm), which is consistent with previous report[57]. For the uniform BFO/STO(001) film, the absorption spectrum is almost the same as pure BFO crystals[58], where a ~580 nm absorption edge is seen, as indicated by a blue arrow. It is of great interest to find that the absorption spectrum for the strain gradient BFO nanostructure is largely modified compared with the BFO/STO(001) film. There is no obvious absorption edge at 580 nm, and the absorption edge is largely extended towards the infrared direction covering much more visible light spectrum, as marked with the red arrow. This phenomenon suggests that the strain gradient induces a continuous bandgap change in the BFO nanostructures (bandgap gradient), which is responsible for the enhancement of visible light absorption since a constant bandgap only induces a specific absorption edge. We note previous studies indicate that chemical gradient also induces bandgap gradient and enhances solar absorption for $TiO_2$ photocatalyst[33]. Moreover, theoretical calculations and nanobeam bending experiments further suggest that strain gradient could introduce bandgap gradient in semiconductors[3,16,17]. Thus our results supply a novel strategy to integrate strain gradient in materials which could be used to modify the band structures of materials and enhance the performance of photocatalysts.

In summary, we have artificially produced a giant linear strain gradient in the $BiFeO_3$/$LaAlO_3$ multilayer nanostructures by a controlled pulsed laser deposition via a high deposition flux mode. Aberration-corrected STEM observation shows that the as-received giant strain gradients are dominated by synergetic interfacial dislocation arrays with $a<100>$ and $a<110>$ Burger vectors. The well aligned $a[001]$ Burger vector components severely rotate the $BiFeO_3$ lattice, result in a long-range giant strain gradient and lead to many exotic properties. The interfacial dislocations herein are very useful modulations other than deleterious ingredients as generally cognized. Our calculations indicate the elastic energy consumption for producing such a giant strain gradient is extremely lower than previously regarded, and our experiments show that the giant strain gradient enables to transfer across a multilayer structure possibly reaching a practical scale. The present results may also stimulate the relevant studies on other epitaxial systems with large lattice mismatch, which are not favoured in the past. Our study provides an opportunity to quantitatively measure the contribution of inhomogeneous strains and assemble them in a nanostructure for the development of novel device concepts, which should involve lead-free electromechanical actuators, high-efficient energy harvesting devices and photocatalysts.

## Methods

**Material preparation.** $BiFeO_3$ nanostructures were deposited by pulsed laser deposition, using a Lambda Physik LPX 305i KrF ($\lambda = 248$ nm) excimer laser. A sintered $BiFeO_3$ ceramic target (3 mol% Bi-enriched) and a stoichiometric $LaAlO_3$ ceramic target were used. The target-substrate distance was 40 mm. The background pressure was $10^{-5}$ Pa. Before deposition, all substrates were pre-heated at 750 °C for 5 min to clean the substrate surface and then cooled down to the growth temperature (10 °C min$^{-1}$). After deposition, the samples were annealed at their growth temperature in an oxygen pressure of $5 \times 10^4$ Pa for 10 min, and then cooled down to room temperature at a cooling rate of ~5 °C min$^{-1}$. Commercial, one-side polishing $LaAlO_3$(001) single-crystal substrates with 10 mm × 10 mm × 0.5 mm dimension were used for film deposition.

**HAADF-STEM imaging and strain analysis.** The samples for the HAADF–STEM experiments were prepared by slicing, gluing, grinding, dimpling and finally ion milling. A Gatan PIPS was used for the final ion milling. Before ion milling, the samples were dimpled down to <20 μm. The final ion milling voltage was <1 kV to reduce ion beam damage. HAADF–STEM images were recorded using aberration-corrected scanning transmission electron microscopes (Titan Cubed 60–300 kV microscope (FEI) fitted with a high-brightness field-emission gun (X-FEG) and double Cs corrector from CEOS, and a monochromator operating at 300 kV). The beam convergence angle is 25 mrad, and thus yields a probe size of <0.10 nm. The diffraction contrast image was recorded using a conventional TEM (Tecnai G2 F30 (FEI) working at 300 kV). Large-scale strain fields were deduced by using custom plugins of GPA under the framework of Gatan DigitalMicrograph software. The visualization of the strains and lattice rotations was carried out using Gatan DigitalMicrograph software.

**Lattice rotations resulting from the interfacial dislocation arrays.** Lattice rotations ($\omega$) are derived from the HAADF–STEM images via GPA, during which $LaAlO_3$ substrate is chosen as the reference lattice. The observed linear strain gradient resulting from the lattice rotation can be persevered in lead-free perovskite

**Elastic energy consideration of perovskite under linear strain gradient.** The elastic energy consumption for producing the present observed strain gradients is calculated. Two types of energies are involved here. One is the elastic energy consumption; and the other is energies of the interfacial dislocation arrays.

**Data availability.** The data that support the findings of this study are available from the corresponding author upon request.

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

## Acknowledgements

This work is supported by the National Natural Science Foundation of China (No. 51231007, 51571197, 51501194, 51671194 and 51521091), National Basic Research Program of China (2014CB921002), and the Key Research Program of Frontier Sciences CAS (QYZDJ-SSW-JSC010). Y.L.T. acknowledges the IMR SYNL-T.S. Kê Research Fellowship and the Youth Innovation Promotion Association CAS (No. 2016177).

## Author contributions

X.L.M. and Y.L.Z. conceived the project of interfacial characterization in oxides by using aberration-corrected STEM. Y.L.T., Y.L.Z. and X.L.M. designed the experiments. Y.L.T. performed the thin-film growth and STEM observations. Y.L. participated in the thin-film growth. Y.J.W. carried out the digital analysis of the STEM data. All authors contributed to the discussions and manuscript preparation.

## Additional information

**Competing interests:** The authors declare no competing financial interests.

