## [Peer Review File · Nature Communications]

Reviewers' comments:

Reviewer #1 (Remarks to the Author):

This is a nice detailed finding that is suitable for publication in Nature Comms. However there are a number of issues that need clarification:

1. The authors must include some discussion of the role of chemical inhomogeneity that may be driven by such large strain gradients. There is enough body of evidence (for example Arredondo et al, Advance Materials 2010,

<http://onlinelibrary.wiley.com/doi/10.1002/adma.200903631/abstract>). For example perhaps EELS mapping of the Fe+3 sites along dislocation lines?

2. It is a really interesting concept here that the dislocations are used to enhance properties. Conventional wisdom is that dislocations deteriorate properties (Nagarajan et al, APL 2005). So the authors must do a better job of enhancing this aspect of their finding.

3. Bottom of page 4- "This gradual increase of...array"- where is the evidence for this claim?

4. If it is high flux that generates this unique interfacial microstructures, then can the authors show it does not happen for low flux case?

5. Figure 1 is very interesting. Can the authors capture data from the orthogonal zone to complete the 3-D picture?

6. The strain gradient will lead to polarization gradient. Typically this would lead to smearing of polarization (see the papers by Bratkovsky and Levanyuk) but in modern times it shown it can also lead to exotic topological structures. Can the authors perform displacement mapping to see if there any fancy polarization domains at the vicinity of such larg estrain gradients.

7. The authors state "The comparison between a fully....is not affected by..."- This is surprising. I would expect it to matter; consider the role of substrate induced constraint on the domain structure and electromechanical properties of ferroelectric.

8. Some technical details are missing-

-What is the zone axis for Figure 1

-What is the reference strain for the GPA mapping

-Some of the model details need to be included in the main manuscript so that the reader is able

to appreciate how the model was arrived at, without having to turn to supplementary information. This always breaks the flow of reading.

9. Finally the authors mention several times new applications- can they elaborate or give examples of how specifically such large linear strain structures may be useful?

10. I have several English edits and minor questions (mostly related to language) and these are marked on the scanned pdf.

In summary this is a very nice finding , which when modified as per above should be suitable for publication.

Reviewer #2 (Remarks to the Author):

By using high deposition flux, interestingly, the authors discovered giant linear strain gradient in the BFO-based nanostructure, which is attributed to the dislocations observed at the BFO/LAO interface. However, some crucial data and evidence to prove significance of this work are missing! Further linking questions are as follows.

1.The authors mainly claimed that the giant linear strain gradient can be created by using high deposition flux. Is this method applicable to other ferroelectrics, such as PZT and BaTiO₃? And how about quality of the samples created in this way?

2.As the authors well exemplified in the introduction, the inhomogeneous strain may dramatically alter the physical properties of some nano-materials. What is the innovation of the achieved giant strain gradient on property of the multiferroic BFO nanostructure? For example, are there any changes of the electrical, magnetic, piezoelectric and optical properties? The structure-property relation should be established!

3.The authors estimated that the flexoelectric coupling may induce a polarization of several $\mu\text{C}/\text{cm}^2$. Unfortunately, direct experimental evidence relating to this point was missing, although the aberration-correction microscopy is accessible for the authors! How does the strain gradient change the local ferroelectric polarization? Particularly, how does the polarization evolve, both in the in-plane and out-of-plane directions, along the normal direction inside the BFO layer? A precise measurement as documented in Phys. Rev. Lett. 102, 117601 (2009) and Nat. Mater. 7, 57 (2007) is highly recommended to clarify the relation. If STEM imaging was used, the artificial effects of (unavoidable) small specimen tilting, astigmatism and coma should be carefully removed by means of image simulations [J. Electron. Microsc. 61, 207 (2012)].

4.How does the strain gradient evolve as a function of repeating sequence of the LAO and BFO layers? Clearly, the results presented in Fig. 4c and 4d are not sufficient and not rigorous. Furthermore, is the LAO layer also polarized by the giant linear strain gradient near the interface [Phys. Rev. B 79, 081405(R) (2009)]?

5.The authors present two cases for bending deformation in Fig. 1 and Fig. S5, S6. How is the growth relation between these two cases in this nanostructure? Do they grow in a face-to-face and/or back-to-back relation, or the distribution is completely random? How is the statistical results of the achieved strain gradient with respect to density and relation of the two types of dislocations? In addition, the high-resolution HAADF images were of low quality. Can they be as good as in your another paper (Ref. 10)?

Although the strain gradient created in this way is interesting, practicability of the method and its influence on quality of the samples should be further confirmed. Furthermore, the structure-property relation should be established to highlight importance of this discovery, as exemplified in Ref. 30. My overall feeling is negative, and I cannot accept publication of the paper in the top-level

journal of Nature Communications. Maybe a chance of major revision deserves, then I'll see whether sufficient experimental evidences can be presented.

Reply to referees' comments:

Reply to referee #1:

We appreciate the positive evaluations by the referee that "This is a nice detailed finding that is suitable for publication in Nature Communications", and that "It is a really interesting concept here that the dislocations are used to enhance properties".

In the meanwhile, the referee also raises several concerns which should be further clarified. We address all these questions one-by-one in the following.

(1) The authors must include some discussion of the role of chemical inhomogeneity that may be driven by such large strain gradients. There is enough body of evidence (for example Arredondo et al, Advance Materials 2010). For example, perhaps EELS mapping of the Fe+3 sites along dislocation lines?

Reply to Question (1):

Yes, in this work we show an interesting concept that the dislocations are used to enhance properties, and we fully agree with the referee's proposal that large strain gradient near a dislocation core may influence the chemical inhomogeneity around the dislocation core, as well studied in the recommended reference. In our present manuscript, although what we emphasize is that the large strain gradients results from the geometric configurations of a dislocation array, the strain gradient may also induce cation non-stoichiometry and Cottrell atmospheres around dislocation cores. This topic is very well addressed in perovskite ferroelectric films (Arredondo et al, Advance Materials 2010). Indeed, EELS mapping is possible to show atomic and electronic structures at the core of crystallographic defects, such as Ruddlesden-Popper faults in doped BaSnO₃ films, as seen in our recent publication in Scientific Reports (2015, *Atomic mapping of Ruddlesden-Popper faults in transparent conducting BaSnO₃-based thin films*). In this revised manuscript, we made relevant discussions as seen in the first paragraph of page 2.

(2) It is a really interesting concept here that the dislocations are used to enhance properties. Conventional wisdom is that dislocations deteriorate properties (Nagarajan et al, APL 2005). So the authors must do a better job of enhancing this aspect of their finding.

Reply to Question (2):

We appreciate the nice evaluation and advices given by the referee. As suggested, we now enhance this new finding by adding some new experimental data. The UV-visible

absorption spectrums for the present nanostructure array and a uniform BFO film on SrTiO₃(001) substrate are compared, and the new data is shown in figure 6 in the revised manuscript. It can be seen that, compared to the uniform BFO film with an absorption edge of about 580 nm (which is consistent with previous reported values), the present BFO nanostructure array with strain gradient has an extended absorption edge largely toward the infrared side covering more visible light spectrum. This effect indicates that the band structure was probably varied continuously in the BFO nanostructures with strain gradient, since it is demonstrated that strain gradient can also induce bandgap gradient (refs. 3, 16, 17). Particularly, the extended absorption edge suggests that this idea can be used as an enhancement of solar absorption of photocatalysts materials.

(3) Bottom of page 4- "This gradual increase of...array"- where is the evidence for this claim?

Reply to Question (3):

We appreciate the kind reminder -- the description here may lead to some confusing. Since the synergetic effect of the dislocation array is well illustrated in the later part of this manuscript, we remove this sentence here.

(4) If it is high flux that generates this unique interfacial microstructures, then can the authors show it does not happen for low flux case?

Reply to Question (4):

Indeed, we tried low flux case in our study and we found that the BFO/LAO interface is completely coherent, as shown in Fig. R1. In fact, the famous strain driven tetragonal BFO (T BFO) was grown here, as confirmed by the FFT pattern (seen in the inset).

Figure R1. High-resolution HAADF images showing the coherent interface at BFO/LAO grown by PLD with a low deposition flux mode.

(5) Figure 1 is very interesting. Can the authors capture data from the orthogonal zone to complete the 3-D picture?

Reply to Question (5):

Fig. 1 is viewed along [100] direction. In Fig. 1, the bending of the BFO and above LAO lattice occurs along the in-plane direction, which means that if the bending is viewed along the orthogonal zone [010], the BFO and above LAO lattice will not be in zone axis direction when the LAO substrate is exactly on the [010] direction. This will blur the image of the BFO/LAO nanostructures. In fact, we have observed this effect in our experiments, since it is not possible that the bending of BFO lattice is always in the in-plane direction.

(6) The strain gradient will lead to polarization gradient. Typically this would lead to smearing of polarization (see the papers by Bratkovsky and Levanyuk) but in modern times it shown it can also lead to exotic topological structures. Can the authors perform displacement mapping to see if there any fancy polarization domains at the vicinity of such large strain gradients.

Reply to Question (6):

We agree with the referee that the interaction of strain gradient and spontaneous polarization (P_s) of BFO may induce exotic topological structures. However, it is known that the measured flexoelectric coefficients (f) for perovskite are on the order of 10^{-9} – 10^{-8} C/m (refs. 19, 20, 30 in the main text), thus the present strain gradient over 10^6 /m may induce several $\mu\text{C}/\text{cm}^2$ flexoelectric polarizations (P_f) as estimated according to $P_f = f \times \Delta S$. This value is smaller when compared with the spontaneous polarization (P_s) of BFO (around $90 \mu\text{C}/\text{cm}^2$), thus it is very difficult to identify these small, strain gradient induced polarizations by displacement mapping since it is usually submerged by the P_s of BFO. We have tried this method and only the P_s of BFO can be detected.

(7) The authors state “The comparison between a fully...is not affected by...”- This is surprising. I would expect it to matter; consider the role of substrate induced constraint on the domain structure and electromechanical properties of ferroelectric.

Reply to Question (7):

We fully agree with the referee that the substrate-induced constrain affects the domain structure and the electromechanical properties of ferroelectrics (as reported in *Nat. Mater.* **2**, 43–47 (2003)), and the dimension influences the property of ferroelectrics. However, in our present study we are comparing the elastic energy differences between a fully strained and a strain gradient state, by theoretical calculations. In fact, the elastic energy differences here refer the energy density differences (per unit volume), which are not affected by the in-plane dimensions in the theoretical calculations.

(8) Some technical details are missing- -What is the zone axis for Figure 1 -What is the reference strain for the GPA mapping. -Some of the model details need to be included in the main manuscript so that the reader is able to appreciate how the model was arrived at, without having to turn to supplementary information. This always breaks the flow of reading.

Reply to Question (8):

We appreciate these suggestions and now we provide all these information in the revised manuscript and revised figures. The zone axis is added in the figure caption of Fig. 1, which is the [100] direction. The LaAlO₃ substrate is chosen as the reference lattice for the GPA mapping, which is indicated in the figure caption of Fig. 2. The schematic model indicating the elastic energy calculation involving the strain gradient is now added in Fig. 5(a) as an inset.

(9) Finally the authors mention several times new applications- can they elaborate or give examples of how specifically such large linear strain structures may be useful?

Reply to Question (9):

As replied in Question (2) and shown in Fig.6, such a large strain gradient may be used to modulate bandgap of materials, and here the strain gradient induced continuous bandgap adjustment can be potentially used to enhance the solar absorption of materials that can be used as photocatalysts.

(10) I have several English edits and minor questions (mostly related to language) and these are marked on the scanned pdf.

Reply to Question (10):

Many thanks must be given to the referee for the language editing. Almost all of the suggested revisions are accepted in the new manuscript.

Reply to referee #2:

We appreciate the positive evaluations by the referee that “the strain gradient created in this way is interesting”. Nevertheless, the main concern raised by the referee is that the structure-property relation should be further established.

We appreciate the constructive suggestions raised by the referee, and in this revised manuscript, we show the structure-property relationship by adding some new experimental data. We show that the UV-visible absorption range in the strain gradient BFO nanostructures are much enlarged, compared with that of the uniform BFO film. The details of the data are shown in Fig. 6 in the revised manuscript, which indicates the potential utility of the strain gradient in perovskite oxide as photocatalysts.

In addition, we now address all of the detailed concerns one-by-one in the following.

(1) The authors mainly claimed that the giant linear strain gradient can be created by using high deposition flux. Is this method applicable to other ferroelectrics, such as PZT and BaTiO₃? And how about quality of the samples created in this way?

Reply to Question (1):

Indeed, in our present study we find that the high deposition flux is of critically important for getting the giant linear strain gradient. Actually, we tried low flux case in our study and we found that the BFO/LAO interface is completely coherent and the famous strain-driven tetragonal BFO (T BFO) was grown (seen in the reply (4) to the first referee).

So, in our present study we show a novel approach to engineer the strains in perovskite oxide films. As seen in our manuscript, it is possible to preserve such strain gradient in the BFO nanostructure; and particularly, by growing other kind of perovskite oxide on top of the BFO nanostructure, such as LAO here, it is feasible to transfer this strain gradient to the upper LAO. By this guide, it is also possible to introduce the strain gradient in other kind of perovskite oxide, such as PZT and BaTiO₃. We checked plenty of the BFO nanostructures, the sample quality is pretty good with sharp interface, as also indicated by our UV-visible absorption tests.

(2) As the authors well exemplified in the introduction, the inhomogeneous strain may dramatically alter the physical properties of some nano-materials. What is the innovation of the achieved giant strain gradient on property of the multiferroic BFO nanostructure? For example, are there any changes of the electrical, magnetic, piezoelectric and optical properties? The structure-property relation should be established!

Reply to Question (2):

As suggested, we now add a new experiment evidence to get better understanding of the structure-property relationships. We have performed UV-visible absorption measurements on the strain gradient BFO nanostructures, which are further compared with that of a uniform BFO film on SrTiO₃(001) substrate. We add a new figure (Fig.6) to show these data in the revised manuscript (also seen in the following).

Figure R2. UV-visible absorption measurements on the LaAlO₃/BiFeO₃/LaAlO₃(001) nanostructures. A uniform BiFeO₃/SrTiO₃(001) film with about 30 nm thickness, a bare LaAlO₃(001) substrate and a bare SrTiO₃(001) (STO(001)) substrate were also measured here for comparison.

It is seen that, compared to the uniform BFO film with an absorption edge of about 580 nm (which is consistent with previous reported values), the BFO nanostructure array with giant strain gradient has an extended absorption edge largely toward the infrared side covering more visible light spectrum. This effect indicates that the band structure is probably varied continuously in the BFO nanostructures with giant strain gradient, since it was demonstrated that strain gradient may also induce bandgap gradient (refs. 3, 16, 17). Particularly, the extended absorption edge suggests that this idea can be used as an enhancement of solar absorption of materials that could be used as photocatalysts.

(3) The authors estimated that the flexoelectric coupling may induce a polarization of several $\mu\text{C}/\text{cm}^2$. Unfortunately, direct experimental evidence relating to this point was missing, although the aberration-correction microscopy is accessible for the authors! How does the strain gradient change the local ferroelectric polarization? Particularly, how does the polarization evolve, both in the in-plane and out-of-plane directions, along the normal direction inside the BFO layer? A precise measurement as documented in Phys. Rev. Lett. 102, 117601 (2009) and Nat. Mater. 7, 57 (2007) is highly recommended to clarify the relation. If STEM imaging was used, the artificial effects of (unavoidable) small specimen tilting, astigmatism and coma should be carefully removed by means of image simulations [J. Electron. Microsc. 61, 207 (2012)].

Reply to Question (3):

We thank the referee for these constructive suggestions and the classic references. Indeed, aberration-correction microscopy is an effective way to study local polarization evolution in ferroelectrics. Nevertheless, here we estimate the flexoelectric effect by using the strain gradient which is directly obtained by experiment. It is known that the measured flexoelectric coefficients (f) for perovskite are on the order of 10^{-9} – 10^{-8} C/m (refs. 19, 20, 30 in the main text). Thus the present strain gradient over $10^6/\text{m}$ may induce several $\mu\text{C}/\text{cm}^2$ flexoelectric polarizations (P_f) as estimated according to $P_f = f \times \Delta S$. It is noted that this value is small when compared with the spontaneous polarization (P_s) of BFO (around $90 \mu\text{C}/\text{cm}^2$). Thus it must be very difficult to identify these small, possible strain gradient induced polarization components since it is usually submerged by the P_s of BFO. We have tried this method and only the P_s of BFO can be detected.

We agree that a precise measurement of the P_s evolution is important for understanding the fine polarization behaviors. The methods and the advantages in all of the above mentioned literatures are cited in this revised manuscript, although the main claim in our present study is the novel strategy to preserve strain gradient in perovskite oxide as potential device concept, such as flexoelectric based ferroelectric functional interfaces, bandgap modulations and photocatalysts.

(4) How does the strain gradient evolve as a function of repeating sequence of the LAO and BFO layers? Clearly, the results presented in Fig. 4c and 4d are not sufficient and not rigorous. Furthermore, is the LAO layer also polarized by the giant linear strain gradient near the interface [Phys. Rev. B 79, 081405(R) (2009)]?

Reply to Question (4):

We appreciate the constructive discussions and the fantastic references given by the referee. Fig. 4 indicates the great possibility that the strain gradient can be preserved if the BFO/LAO nanostructures are repeatedly constructed. It is worthwhile to mention that strict conditions must be explored and optimized in order to precisely control the growth of perfect multilayer BFO/LAO nanostructures.

For the polar behavior of the LAO layer, as replied above, it is polarized through the flexoelectric coupling: $P_f = f \times \Delta S$, since the strain gradient in LAO is nearly the same as that in BFO (Fig. 2d). The flexoelectric coefficients (f) for perovskite are on the order of 10^{-9} – 10^{-8} C/m (refs. 19, 20, 30 in the main text). Thus the present strain gradient over $10^6/\text{m}$ may induce several $\mu\text{C}/\text{cm}^2$ flexoelectric polarizations (P_f).

(5) The authors present two cases for bending deformation in Fig. 1 and Fig. S5, S6. How is the growth relation between these two cases in this nanostructure? Do they grow in a face-to-face and/or back-to-back relation, or the distribution is completely random? How is the statistical results of the achieved strain gradient with respect to density and relation of the two types of dislocations? In addition, the high-resolution HAADF images were of low quality. Can they be as good as in your another paper (Ref. 10)?

Reply to Question (5):

We thank the referee for these detailed discussions. As indicated by the referee, we have carefully checked plenty of cross-sectional images, and we found that all face-to-face, back-to-back and face-to-back relations exist. These nanostructures tend to distribute randomly. For the 'statistical results of the achieved strain gradient with respect to density and relation of the two types of dislocations statistical results', we note that the density of dislocation is almost constant (spacing is 6-8 nm), as the in-plane HAADF image indicated (Fig. S3). The $a\langle 100 \rangle$ and $a\langle 110 \rangle$ dislocations coexist but they are distributed nonuniformly, thus the giant strain gradient is induced. If there is only $a\langle 100 \rangle$ or $a\langle 110 \rangle$ dislocations, it is obviously that the strain gradient cannot be obtained.

We are grateful to the referee for the nice comment on our previous paper (Ref.10). The low-magnification HAADF images in the present paper are likely of low-quality compared with the atomic images we showed earlier (Ref.10), they do contain much information. To apply GPA method and extract strains based on a HAADF image, we should emphasize that lower-magnifications images are much better. And, such a combination of HAADF imaging and GPA method has various advantages: it is straightforward, accurate on the unit-cell scale, relatively insensitive to crystal orientation and therefore helpful for large-scale (Ref. 40).

Reviewers' comments:

Reviewer #1 (Remarks to the Author):

The authors did a nice job answering the referees' major concerns. The only revision I ask is that authors include the low flux data as supplementary so that the story is fully complete. Following that the paper is ready for acceptance.

Reviewer #2 (Remarks to the Author):

Although the authors added the UV-visible absorption measurements, my major concerns about reproducibility and importance of the linear strain gradient were not dispelled by the authors, who didn't provide more substantial experimental evidences and results to consolidate and enhance quality of this paper. Typically:

(1) The highlight of this paper is material design, which indicates that this methodology should be applicable to other material system, e.g., PZT. However, adaptability of this methodology over other material systems was not verified and reproduced, which questions the practicability and its potential applications claimed by the authors.

(2) BiFeO₃ is a typical paradigm in which the influence of strain gradient on the physical (ferroelectric and magnetic) properties can be easily and deserves to be well established. However, the authors didn't present any of their endeavor in clarifying relations of these aspects, even the polarization changes (e.g., over areas with distinct strain gradient) quantified by HAADF-STEM, which is fully accessible by the authors.

(3) It has to be mentioned that the analysis of GPA is unit-cell scale accuracy, rather than atomic-scale. The flexoelectric polarization can be estimated in that way, but direct measurement is absolutely needed.

Overall, the authors didn't show their proper effort in improving quality of this paper, but leaving so many pending and unclarified questions. Therefore, I reject to publish this paper in the high-level journal of Nature Communications.

Reviewer #3 (Remarks to the Author):

This manuscript has successfully demonstrated a nonlocal linearly strain-modulated BiFeO₃ nanostructure with a giant built-in strain gradient as high as 106 per meter. By taking the UV-visible absorption measurements, a continuous bandgap change that generated from the inhomogeneous built-in strain has been confirmed, which has proved the possibility that large linear strain gradient can effectively affect the macroscopic property of the lead-free perovskite nanostructures.

This manuscript has also demonstrated two important techniques: one is using the PLD deposition method to build giant linear strain gradient in the BiFeO₃/LaAlO₃ multilayer nanostructures; and the other is using aberration corrected STEM to directly detect the type and distribution of the dislocations especially near the interface and further using GPA method to extract long-range complex and inhomogeneous strain distribution. These techniques are of critical importance and show great advantages over other techniques. Authors have provided detailed experimental procedure and parameters which will help the other researchers continue applying these film-growth and characterization techniques into other materials systems and nanostructures to develop exotic properties and novel devices.

This manuscript has complete and fluent logic flow and easy-to-understand statements. Thus, I suggest this manuscript to be accepted with minor modification. The questions that need to be addressed or noticed are as following.

1. There should be more explanation for this sentence "The comparison between a fully 2D strained and a linear strained BFO ... is not affected by the in-plane dimensions" in the main text.
2. Some spaces are missing, like "(8.854×10⁻¹²F/m)".
3. Could authors do some experiments to further confirm the approximate value of the flexoelectricity-driven electric field (E_f)? Since authors have declared a large built-in electric field exceeding 1 MV/m in the Abstract part, I believe many readers will get interested at this point after glancing over this abstract and want to get more confirmed answers in the main text.
4. In the Method section and the materials part, the BFO and LAO targets are mentioned, however the description of the LAO substrate has gone missing.
5. There are some strange symbols at the end of some references, like ref. 22 and ref. 31.
6. Authors should provide some explanation of the reason for further depositing LAO on the top of the BFO nanostructures at the very beginning of the main text (especially around the interpretation of Fig. 1 and Fig. 2), although the reason is somehow obvious.
7. At the end of the supplementary materials, I am wondering where was the number "~24nm" derived from?

Reply to referees' comments:

Reply to referee #1:

We appreciate the evaluations by the referee that "The authors did a nice job answering the referees' major concerns"; we also appreciate the recommendation that "the paper is ready for acceptance" after "including the low flux data as supplementary so that the story is fully complete".

We have now added a supplementary figure (Fig.S3) in the Supplementary Materials showing the low flux BFO/LAO case. It is shown that only strain-driven tetragonal BFO phase (T phase) is observed and the BFO/LAO interface is free of interfacial dislocations.

Reply to referee#2:

(1) The highlight of this paper is material design, which indicates that this methodology should be applicable to other material system, e.g., PZT. However, adaptability of this methodology over other material systems was not verified and reproduced, which questions the practicability and its potential applications claimed by the authors.

Reply to Question (1):

In our present study, the first BFO layer is the prototype of the idea of integrating linear strain gradient into oxide nanostructures. As is seen in our manuscript, the lead free LAO layer, successively grown on BFO, also displays the linear strain gradient which is the same as that in the first BFO layer. In other words, the critical condition for introducing such a linear strain gradient in perovskite oxides only depends on the first BFO layer. Following this methodology, once the linear strain gradient is generated in the first BFO layer via high flux deposition, the linear strain gradient is able to automatically transfer into the next perovskite layer epitaxially grown on BFO via common growth condition. These perovskite oxides should include LAO, PZT, BaTiO₃ *etc.*, regardless of their mismatches with BFO.

The main reason for the transfer of such a linear strain gradient is that the perovskite oxide layer via the common growth mode on BFO is relaxed only via the *a*[010] type misfit dislocations (without an out-of-plane Burgers component, so the out-of-plane bending remains), which has been proved in the LAO layer and next BFO layer of our present study.

(2) BiFeO₃ is a typical paradigm in which the influence of strain gradient on the physical (ferroelectric and magnetic) properties can be easily and deserves to be well established. However, the authors didn't present any of their endeavor in clarifying relations of these aspects, even the polarization changes (e.g., over areas with distinct strain gradient) quantified by HAADF-STEM, which is fully accessible by the authors.

Reply to Question (2):

As far as PTO and BFO are considered, indeed effect of strains on physical property is well known. In contrast, *strains gradients* on physical property are less established.

The strain gradient-related physical properties, especially the flexoelectric responses of ferroelectrics/dielectrics have received great attention in recent years since the nanoscale flexoelectricity is expected to offer many advantages over other technologies, such as its generality and linear/non-hysteretic essence (ref. 19 in the main text). However, studies in this area are still in its infancy and many puzzles still remain (ref. 19 in the main text). These puzzles include how to precisely measure flexoelectric coefficients of materials and how to systematically integrate the strain gradient into nanodevices.

In case of BiFeO₃, the influence of strains on the physical (ferroelectric and magnetic) properties is well established, but strain gradients on the physical properties are largely unknown; the main obstacle herein is how to systematically integrate the strain gradient into BiFeO₃. To the best of our knowledge, except for the discussions of strain gradient-related properties at *domain walls* [refs. 30, 31 in the main text], there is very few work on integrating of linear strain gradient into BiFeO₃ nanostructures, on strain mapping, and on structure-property relationships.

Indeed, we have performed ferroelectric ion displacement mappings and we find that the strain gradient-induced polarization change in BFO unit cells is almost undetectable along the out-of-plane direction, as is seen in Figure R1 below. Basically, the Fe ion displacement is still along the quasicubic <111> direction of the BFO unit cells, which is almost the same as the unstrained BFO unit cells. This observation indicates that the strain gradient-induced polarization changing in the BFO lattice by the strain gradient here is indeed not big, when compared with the spontaneous polarization of the BFO itself. This result is consistent with our estimation based on the flexoelectric effect, which is obtained by using the strain gradient measured in our experiment. For example, it is known that the measured flexoelectric coefficients (f) for perovskite are on the order of 10^{-9} – 10^{-8} C/m (refs. 19, 20 30 in the main text). The present strain gradient over 10^6 /m may induce several $\mu\text{C}/\text{cm}^2$ flexoelectric polarizations (P_f) as estimated according to $P_f = f \times \Delta S$. This value is indeed very small when compared with the spontaneous polarization (P_s) of BFO (around $90 \mu\text{C}/\text{cm}^2$).

Figure R1. Fe ion displacement mapping in the first BFO layer. (a), High resolution HAADF-STEM imaging of the BFO layer. (b), Fe ion displacement mapping based on (a).

(3) It has to be mentioned that the analysis of GPA is unit-cell scale accuracy, rather than atomic-scale. The flexoelectric polarization can be estimated in that way, but direct measurement is absolutely needed.

Reply to Question (3):

Within the scope of TEM method, we believe that the strain gradient-based estimation is more suitable for calculating the flexoelectric polarization (P_f), since the P_f is in the order of only several $\mu\text{C}/\text{cm}^2$. This value is obviously much smaller than the spontaneous polarization (P_s) of BFO (around $90 \mu\text{C}/\text{cm}^2$). As replied in question (2), we have tried direct measurement by using high-magnified HAADF-STEM images, and we find that the flexoelectric polarization is indeed much smaller and cannot be immediately captured by direct measurement.

Up to date, very few groups in the world are able to perform this kind of measurement except that Prof. Catalan's group in Spain tried to measure the flexoelectric responses of a system which is designed to have a special geometric shape. Nevertheless, this method cannot be readily used to our self-integrated linear strain gradient. In addition, such kind of measurement does not in any way influence the overall physics of our present paper.

Reply to referee#3:

We appreciate the positive evaluations by the referee who precisely highlights the two important techniques: one is using the PLD deposition method to build giant linear strain gradient in the BiFeO₃/LaAlO₃ multilayer nanostructures; and the other is using aberration corrected STEM to directly detect the type and distribution of the dislocations especially near the interface and further using GPA method to extract long-range complex and inhomogeneous strain distribution. We fully agree with the referee that "Authors have provided detailed experimental procedure and parameters which will help the other researchers continue applying these film-growth and characterization techniques into other materials systems and nanostructures to develop exotic properties and novel devices", and that "These techniques are of critical importance and show great advantages over other techniques".

In addition, the referee raises several issues suggested for minor modifications. Here we address all these concerns one-by-one in the following.

(1) There should be more explanation for this sentence "The comparison between a fully 2D strained and a linear strained BFO ... is not affected by the in-plane dimensions" in the main text.

Reply to Question (1):

We appreciate the referee for this kind reminder. Actually here we wanted to compare the elastic energy of a BFO island under two different strain states. Thus, for a calculation of elastic energy, the in-plane dimensions of just one BFO island can be arbitrarily set as any value, which does not affect the comparison result.

The question raised by the referee reminds us that the previous expression in the manuscript may results in some misleading, we now have revised it as "To compare the elastic energy of a BFO island in a fully 2D strain state with that in a linear strain state, we arbitrarily set $l = t = 100\text{nm}$ in our calculations."

(2) Some spaces are missing, like "(8.854×10⁻¹²F/m)".

Reply to Question (2):

These formats are carefully checked and revised.

(3) Could authors do some experiments to further confirm the approximate value of the flexoelectricity-driven electric field (E_f)? Since authors have declared a large built-in electric field exceeding 1 MV/m in the Abstract part, I believe many readers will get interested at this point after glancing over this abstract and want to get more confirmed answers in the main text.

Reply to Question (3):

This suggestion is very constructive. Actually, in-situ method based on off-axis electron holography in a TEM is a potential way to estimate the electrostatic potentials and electric fields. However, the flexoelectricity-driven electrostatic potentials and electric fields measurements must be strongly correlated with the spontaneous

polarization of BFO. The spontaneous polarization also generates big electrostatic potentials and electric fields inside a BFO film/at the interface. Thus, to separate a flexoelectricity-driven component has been a big challenge. Instead, the calculation approach, as we used in this study, is generally used for estimating the flexoelectricity-driven electric field in ferroelectrics [R1,R2,R3]. Nevertheless, we will pay close attention to how to directly measure the electrostatic potentials and electric fields resultant from this novel strain gradient state, although this requires completely new experiment design including sample preparations and in-situ TEM observation.

(4) In the Method section and the materials part, the BFO and LAO targets are mentioned, however the description of the LAO substrate has gone missing.

Reply to Question (4):

We have added the LAO substrate description in the Method section as "Commercial, one-side polishing LaAlO₃(001) single crystal substrates with 10 mm × 10 mm × 0.5 mm dimension were used for film deposition. "

(5) There are some strange symbols at the end of some references, like ref. 22 and ref. 31.

Reply to Question (5):

These problems may arise from some format collisions of the WORD software. These formats were carefully checked and revised.

(6) Authors should provide some explanation of the reason for further depositing LAO on the top of the BFO nanostructures at the very beginning of the main text (especially around the interpretation of Fig. 1 and Fig. 2), although the reason is somehow obvious.

Reply to Question (5):

We really appreciate this reminder, this revision will further facilitate the readers for understanding the manuscript. "The objective of the LAO/BFO film design here is to study the strain interactions and transfers among the lead-free multilayer structures and corresponding strain induced physical effects." were added at the first paragraph of the Result section.

(7) At the end of the supplementary materials, I am wondering where was the number "~24nm" derived from?

Reply to Question (7):

The thickness of the first BFO layer is about 24nm, which can be seen in Fig. 2(d). To eliminate the possible confusion, we added a supplementary description in the revised manuscript as ".....24 nm (Fig. 2(d) in the main text)".

References in this reply to reviewers:

[R1] D. Lee, et al. Giant Flexoelectric effect in ferroelectric epitaxial thin films. Phys. Rev. Lett. 107, 057602 (2011).

[R2] R. Resta, Towards a Bulk Theory of Flexoelectricity. Phys. Rev. Lett. 105, 127601 (2010).

[R1] A. Gruverman, et al. Mechanical stress effect on imprint behavior of integrated ferroelectric. Appl. Phys. Lett. 83, 728 (2003).

REVIEWERS' COMMENTS:

Reviewer #3 (Remarks to the Author):

The authors have addressed my previous concerns; thus I recommend publication as is.